# Caffeic Acid Phenethyl Ester (CAPE): Biosynthesis, Derivatives and Formulations with Neuroprotective Activities

**DOI:** 10.3390/antiox12081500

**Published:** 2023-07-27

**Authors:** Rebeca Pérez, Viviana Burgos, Víctor Marín, Antoni Camins, Jordi Olloquequi, Iván González-Chavarría, Henning Ulrich, Ursula Wyneke, Alejandro Luarte, Leandro Ortiz, Cristian Paz

**Affiliations:** 1Laboratory of Natural Products & Drug Discovery, Center CEBIM, Department of Basic Sciences, Faculty of Medicine, Universidad de La Frontera, Temuco 4780000, Chile; perezcolladorebeca@gmail.com (R.P.); victor.marinmosi.bq@gmail.com (V.M.); 2Departamento de Ciencias Biológicas y Químicas, Facultad de Recursos Naturales, Universidad Católica de Temuco, Rudecindo Ortega, Temuco 4780000, Chile; viviburgossalgado@gmail.com; 3Department of Pharmacology, Toxicology and Therapeutic Chemistry, Faculty of Pharmacy and Food Sciences, Universitat de Barcelona, 08028 Barcelona, Spain; camins@ub.edu; 4Institut de Neurociències (UBNeuro), Universitat de Barcelona, 08028 Barcelona, Spain; 5Biomedical Research Networking Centre in Neurodegenerative Diseases (CIBERNED), 28031 Madrid, Spain; 6Department of Biochemistry and Physiology, Faculty of Pharmacy and Food Sciences, Universitat de Barcelona, 08028 Barcelona, Spain; jordiolloquequi@ub.edu; 7Institute of Biomedical Sciences, Faculty of Health Sciences, Universidad Autónoma de Chile, Talca 3460000, Chile; 8Departamento de Fisiopatología, Facultad de Ciencias Biológicas Universidad de Concepción, Concepción 4030000, Chile; ivancsbiologicas@gmail.com; 9Department of Biochemistry, Instituto de Química, Universidad de São Paulo, Av. Prof. Lineu Prestes 748, São Paulo 05508-000, SP, Brazil; henning@iq.usp.br; 10Facultad de Medicina, Universidad de Los Andes, Santiago 111711, Chile; uwyneken@uandes.cl (U.W.);; 11Center of Interventional Medicine for Precision and Advanced Cellular Therapy (IMPACT), Santiago 7620001, Chile; 12Instituto de Ciencias Químicas, Facultad de Ciencias, Universidad Austral de Chile, Casilla 567, Valdivia 5110566, Chile; leandro.ortiz@uach.cl

**Keywords:** caffeic acid phenethyl ester, CAPE derivatives, anti-inflammatory, NF-κB, antioxidant, Nrf2, neuroprotection

## Abstract

Neurodegenerative disorders are characterized by a progressive process of degeneration and neuronal death, where oxidative stress and neuroinflammation are key factors that contribute to the progression of these diseases. Therefore, two major pathways involved in these pathologies have been proposed as relevant therapeutic targets: The nuclear transcription factor erythroid 2 (Nrf2), which responds to oxidative stress with cytoprotecting activity; and the nuclear factor NF-κB pathway, which is highly related to the neuroinflammatory process by promoting cytokine expression. Caffeic acid phenethyl ester (CAPE) is a phenylpropanoid naturally found in propolis that shows important biological activities, including neuroprotective activity by modulating the Nrf2 and NF-κB pathways, promoting antioxidant enzyme expression and inhibition of proinflammatory cytokine expression. Its simple chemical structure has inspired the synthesis of many derivatives, with aliphatic and/or aromatic moieties, some of which have improved the biological properties. Moreover, new drug delivery systems increase the bioavailability of these compounds in vivo, allowing its transcytosis through the blood-brain barrier, thus protecting brain cells from the increased inflammatory status associated to neurodegenerative and psychiatric disorders. This review summarizes the biosynthesis and chemical synthesis of CAPE derivatives, their miscellaneous activities, and relevant studies (from 2010 to 2023), addressing their neuroprotective activity in vitro and in vivo.

## 1. Introduction

The current demographic shift has led to a surge in the prevalence of diseases affecting older adults, including neurodegenerative disorders. Among them, Alzheimer’s disease (AD) is the first cause of dementia and a significant global public health problem, estimated to affect about 131.5 million people worldwide, with an annual incidence of 4 to 6 million new cases [1]. Furthermore, the second neurodegenerative disorder is Parkinson’s disease (PD), affecting one in every hundred people over 60 years of age. It is estimated that by 2030 there will be 9 million patients with idiopathic PD worldwide [2]. Moreover, psychiatric disorders with increasing worldwide prevalence, such as major depressive disorder (MDD), are relevant and potentially modifiable risk factors for dementia. Neurodegenerative disorders are characterized by excessive damage of key brain structures, which is followed by neuronal function loss, structural alterations, and decrease of cellular survival [3]. However, despite our increasing knowledge about their pathogenic mechanisms, successful therapeutic approaches to tackle neurodegeneration have remained highly elusive.

A wealth of evidence has indicated that oxidative stress and neuroinflammation may have a crucial role in the pathogenesis of neurodegenerative disorders, as they are associated with complex systems that activate programmed cell death cascades [4,5,6,7].

Activation of the Nrf2 and NF-κB pathways is expected to have a prominent role in the control of oxidative stress and inflammation. The nuclear erythroid transcription factor 2 (Nrf2) responds to oxidative stress, mediating cytoprotection in mammalian cells by the production of a set of antioxidant genes called phase II genes, which produce phase II proteins associated with ROS stabilization [8,9,10]. In turn, NF-κB controls pro-inflammatory gene expression. The synthesis of cytokines such as the tumor necrosis factor alpha (TNF-α), interleukin (IL)-1β, IL-6, and IL-8 is directly mediated by NF-κB, as well as the expression of cyclooxygenase-2 (COX2) [11].

To date, conventional single drug-based therapeutic approaches for addressing neurodegenerative disorders have not been entirely satisfactory. For instance, in the pharmacological treatment of AD, since the 1990s, only three small molecules as cholinesterase inhibitors, including galantamine, rivastigmine, and donepezil, together with memantine, a glutamatergic antagonist, are available, alongside two antibodies, aducanumab and lecanemab. Therefore, targeting multiple pharmacological pathways such as inflammation and oxidative stress holds great promise for increasing the likelihood of obtaining potent neuroprotective effects. Among the novel and emerging therapeutic approaches, consuming functional foods enriched with natural bioactive molecules has shown great potential as a palliative or preventative measure for the middle-aged population. For instance, propolis is considered a food supplement with relevant therapeutic properties, including antidiabetic, anticancer or antibacterial activities. Propolis is a large mixture of polyphenols, essential oils, and waxes produced by plants as resinous secretions. This is collected by honeybees, which use it as a waterproof substance to seal hive cracks or holes [12]. Furthermore, bees use propolis as an antiseptic material to prevent infections due to its powerful antibacterial [13] and antiparasitic properties [14]. The large molecular variety of propolis depends on many factors, such as location, the season, and plants distributed around the hive [15]. Despite this, propolis always shows antioxidant, anti-inflammatory, bactericidal, and antifungal properties that could be explained due to the high content and variety of flavonoids with a particular chemical composition [16,17,18].

## 2. Natural Sources of Caffeic Acid Phenethyl Ester

Caffeic acid phenethyl ester (CAPE) is found mainly in propolis. It is a natural polyphenol from the phenylpropanoid family, formed by the esterification of caffeic acid and phenethyl alcohol (Figure 1) [19,20,21].

CAPE is not the major component of propolis, but it is one of the most potent antioxidants supplied by the caffeic acid moiety, which has a higher antioxidant capacity than related phenylpropanoids such as ferulic acid and p-coumaric acid [22]. The concentration of CAPE in propolis varies greatly, from 0 to 11 mg/g of propolis collected in different regions of Turkey [23]. Quantification by high-performance liquid chromatography (HPLC) is a suitable method for its determination [24]. Table 1 shows the concentration of CAPE in some sources. 

The presence of CAPE and caffeic acid was analyzed in 25 species of mushrooms. CAPE was not found in any any of them, unlike caffeic acid, which was quantified in seven mushrooms, including *B. edulis*, *S. commune*, *F. velutipes*, *Agrocybe aegerita*, *P. eryngii*, *P. cystidiosus* and *P. adiposa* in concentrations from 0.004 to 0.577 mg g^−1^ [28].

## 3. Biosynthesis of CAPE

CAPE displays a broad spectrum of beneficial activities, including antitumor activity by inducing apoptosis in many cancer cell lines [26,29,30,31]. For example, on human leukemia HL-60 cells, CAPE inhibits DNA, RNA, and protein synthesis with an IC_50_ of 1.0 μM, 5.0 μM, and 1.5 μM, respectively [32]. Moreover, CAPE induces anti-inflammatory and immunomodulatory responses. These activities have been explained by the activation of Nrf2 and the inhibition of NF-κB. Thus, CAPE is commercially available as an NF-κB inhibitor for in vitro and in vivo studies [33].

This review provides an in-depth examination of the biosynthesis and chemical synthesis of CAPE derivatives, as well as their various activities, with a particular focus on their effects on the Nrf2 and NF-κB signaling pathways implicated in oxidative and inflammatory damage underlying neurodegeneration. 

CAPE is naturally synthesized by the phenylpropanoid pathway, beginning with the amino acids phenylalanine or tyrosine, which produce 4-coumaric acid. The biosynthesis of CAPE is summarized in Figure 2. 

Coenzyme A and 4-coumaric acid are bound by 4-coumaric acid CoA-ligase (4CL) enzymatic activity, producing 4-coumaroyl CoA with an activated carbonyl. This product is further meta hydroxylated by p-coumarate 3-hydroxylase, producing the intermediate caffeoyl CoA, which is the precursor of caffeic acid by hydrolysis of CoA. In the biosynthesis of CAPE, however, the CoA is substituted in one step with phenethyl alcohol by ester linkage. Phenethyl alcohol is synthesized by two enzymes, an aromatic amino acid decarboxylase such as phenylalanine decarboxylase (PDC), followed by a monoamine oxidase such as phenethylamine oxidase (PEO) [34].

The absorption and metabolism of CAPE have not yet been well studied, but these processes should be like those of caffeic acid or related caffeic acid esters. The absorption and metabolism of the radiolabel [3-^14^C] caffeic acid have been studied in rats, showing that absorption starts in the stomach after 1 h post-ingestion and is quickly absorbed in the first portion of the intestine, obtaining high concentrations in plasma after 2 h, and then is excreted mainly by urine as nine derivatives of type sulfonated and glucuronide, with no accumulation in tissues [35]. CAPE is hydrolyzed after 6 h in rat plasma, producing caffeic acid by a carboxylesterase [36].

Caffeic acid esters, such as chlorogenic and rosmarinic acid, have shown a similar pattern, being excreted by urine 4–8 h post-ingestion, with only 3.3 to 4.0% of total polyphenol content remaining [35,37]. The pharmacodynamic and pharmacokinetics of murine and human organisms are different, however. For instance, CAPE shows more stability in human than in rat plasma, where its concentration decreases quickly, producing caffeic acid and by-products, suggesting that CAPE is hydrolyzed by enzymes present in rat plasma but not in human [36].

## 4. CAPE Derivatives as Novel Bioactive Compounds

The structure of CAPE inspired organic chemists to develop chemical analogs that improve the activities of the original molecule. The synthesis of CAPE derivatives has been improved over the years, from the initial Fisher esterification catalyzed with acids, giving low yields, to the utilization of coupling reagents as DCC, the condensation of acid chlorine derivative or microwave assistance. Here we summarize some methods that produce acceptable yields and are easy to apply (Figure 3). 

Choi et al. (2019) produced ester derivatives by reflux overnight with starting alcohol and 1.2 equivalents of Meldrum’s acid, obtining the corresponding malonic acid monoester. Next, 1 eq. of an aldehyde produces the corresponding CAPE derivative by Knoevenagel condensation, as shown in Figure 3, entries 1 to 3 [38]. An alternative pathway involves microwave assistance, considerably reducing reaction times but giving moderate yields (entries 4 to 6).

The N,N′dicyclohexylcarbodiimide (DCC) coupling method was used by Chen et al. in the synthesis of CAPE. Starting with caffeic acid and 5 equivalents of phenethyl alcohol, a 38% yield of CAPE was produced after purification (Figure 4, entry 1). Nagaoka synthetized twenty CAPE analogues by acyl chlorine condensation, starting with caffeic acid and chlorine thionyl in a dry atmosphere. The corresponding alcohol was then added, producing the corresponding CAPE derivative (Figure 4, entry 2a to 2c). The product 4-phenylbutyl caffeate showed antiproliferative activity with EC_50_ of 20 μM against murine colon 26-L5 carcinoma cells [39]. 

Different methods have achieved the synthesis of CAPE and its derivatives. However, some routes were not discussed here because they present low yields, inconvenient reagents, or burdensome purification procedures, unlike the methods summarized in Figure 3 and Figure 4. CAPE derivatives were studied in vitro and in vivo due to the antiviral and anticancer activities displayed. In Figure 5, some of its esters are shown. 

A compound (E)-phenyl 3-(3,4-dihydroxyphenyl) acrylate is the best result of a screening of 19 CAPE-derivatives in a search for inhibitors of the enzyme xanthine oxidase (XO). XO inhibition is a therapeutic strategy for treating hyperuricemia, which results in uric acid crystal accumulation in joints, producing gout or inflammatory arthritis. This CAPE derivative inhibits XO more efficiently than CAPE by π-π interactions with the enzyme. This enzymatic activity, together with the anti-inflammatory capacities of the related molecules, suggests its use in the treatment of hyperuricemia [38]. Moreover, n-Alkyl esters of CAPE have shown antiviral activity against the hepatitis C virus (HCV), with actions depending on the length of the aliphatic chain and substituents in the catechol moiety. Among derivatives with 1, 4, 6, 8, and 10 carbons, octyl caffeate (CAOE, Figure 5) exhibited the highest activity against HCV with an EC_50_ value of 2.7 mM, inhibiting genotype 1b and 2a of the virus, independent of interferon signaling. Moreover, CAOE showed synergistic activity with antivirals such as interferon-alpha 2b, daclatasvir, and VX-222, enhancing their activities. Shen and collaborators evidenced that CAOE is protective against hepato-toxic stimuli and mitochondrial dysfunction induced by tert-butyl hydroperoxide in HepG2 Cells. CAOE showed the best results compared to ten different compounds, including amides and esters of caffeic acid [40,41]. 

A more extended side chain derivative with 12 carbons, decyl caffeate (DC, Figure 5) showed inhibition of human colorectal cancer cells HT-29 and HCT-116 cell proliferation in vivo and in vitro on a xenograft mouse model, enhancing cell-cycle arrest at the S phase via blockade of the STAT3 and Akt signaling pathways. Therefore, DC inhibits cell proliferation by inducing autophagy in HCT-116 colon cells [42].

The replacement of the ester bonding by amide bonding could increase the stability of the molecules in acidic pH and esterase activity, increasing their bioavailability. Thus, many caffeoyl amides have been synthesized (Figure 6), and some of them showed neuroprotective and photoprotective effects in cell and animal models.

Compounds K36 and K36H are closely related to CAPE. They change the ester bonding by amide linkage, showing antiphotodamage properties induced by ultraviolet A (UVA), which is a preponderant factor in skin damage and aging. They were synthesized by the condensation of caffeic acid with 2-phenylethanamine or 2-(4-bromophenyl)ethanamine producing K36 or K36H, respectively. K36 reduces ROS levels via Nrf2, with the consequent antioxidant response mechanisms, such as the expression of the antioxidant enzyme heme oxygenase-1 (HO-1) and the inhibition of metalloproteinase (MMP)-1, MMP-2, IL-6, and PGE2. K36, as well as CAPE, reduces iNOS and COX-2 expression by inhibition of NF-κB [43]. Moreover, K36 reduces the activation of extracellular signal-regulated kinase (ERK) and c-Jun N-terminal kinases (JNK), as well as 8-hydroxy-20-deoxyguanosine (8-OHdG) levels [44]. The p-bromine derivative K36H displays a behavior similar to that of K36, reducing ROS generation, PGE2, NO formation, DNA damage, overexpression of matrix metalloproteinase (MMP)-1, and MMP-2, as well as the expression of Bax and caspase-3, in line with its photoprotective properties. These effects suggest that K36 and K36H could be used as active components of new photoprotector products [45].

The long-chain amides, CAPE dodecyl amide (CAF12) together with CAPE hexyl amide (CAF6), showed neuroprotective effects in an animal model of diabetic retinopathy by ameliorating oxidative stress via increasing the antioxidant enzyme retinal superoxide dismutase (SOD). Moreover, treatment with CAF12 or CAF6 decreased iPF2α levels and AKT phosphorylation in diabetic rats, promoting the survival and growth of retinal neurons. This resulted in the prevention of negative morphological changes in the retina of diabetic rat models treated with streptozotocin (a diabetic inductor drug) compared to rats treated with CAF12 or CAF6 intravitreally for two weeks [46]. Moreover, prior studies with aliphatic amides of caffeic acid with C3, C4, C6, C8, and C12 were evaluated on AD models, showing that CAF12 increases PC12 pheochromocytoma cell survival in serum-deprived conditions, enhancing the nerve growth factor (NGF) effect, and inducing neurite outgrowth via the ERK1/2 and AKT pathways by activation of PI3K. Moosavi and colleagues suggested that these amides could have neuroprotective properties [47]. A CAF12 and CAF6-related amide, denominated compound 13, with a p-methoxy group incorporated into the phenyl moiety, revealed a potent antioxidant and neuroprotective activity with better results than those of 20 other related compounds evaluated [48]. Compound 13 induced memory loss impairment in a transgenic *Drosophila* model, as well as inhibitory or disaggregation activity on Aβ peptides in Alzheimer’s disease (AD) models [49]. Although these amides are structurally related to the esters shown in Figure 3, and possess antiviral and anticancer activities, they have not yet been studied adequately in these disease models. Only a molecule with a shorter side chain, the amide PT-93, was studied in glioblastoma multiforme (GBM), such as T98G, U87, U251 and HT22 cell lines. The compound showed antiproliferative activities in these models by MMP-2 and MMP-9 inhibition [50].

Amides and esters presented here are the result of inducing chemical variations in the phenethyl moiety of CAPE, producing a variety of bioactive compounds. Moreover, Wan et al. (2019) improved the solubility and absorption of CAPE with the synthesis of the 4-O-glucoside of CAPE, called FA-97, by coupling CAPE to an acetyl-protected brominated D-glucose, according to Figure 7 [51].

FA-97 enhances antioxidant and scavenger properties in neuronal cell cultures by directly inhibiting ROS production while also activating Nrf2, which produces an antioxidant cell response. In SH-SY5Y human neuroblastoma and PC12 rat pheochromocytoma cells, FA-97 attenuated H_2_O_2_-induced apoptosis by decreasing ROS and malondialdehyde (MDA) levels, also inducing cellular superoxide (SOD) and glutathione (GSH) dismutase, quinone oxidoreductase 1(NQO1), and HO-1 by activation of the Nrf2 pathway. On AD models, FA-97 reduced learning and memory impairments in mice via reducing neuronal apoptosis [51]. Furthermore, FA-97 proved to be a potential drug against inflammatory bowel disease (IBD), which is an inflammatory intestinal disorder. On a mouse model of colitis employing dextran sodium sulfate (DSS) to induce epithelial damage, FA-97 reduced body weight loss as well as colon damage and length shortening via the activation of HO-1/Nrf2, showing the excellent properties of this compound in gastrointestinal affections [51]. 

## 5. CAPE Inhibits Oxidative Stress by Modulation of the Nrf2 Pathway

The metabolic activity of the brain renders it particularly susceptible to oxidative damage. Indeed, although the brain accounts for only 2% of our body weight, it consumes 20% of the whole body’s oxygen supply. This puts neuronal cells at risk of generating an excess of free radicals that can overwhelm the available antioxidant systems, leading to oxidative stress (OS). Furthermore, OS is associated with the development and progression of different neurodegenerative disorders, including Alzheimer’s disease (AD), Parkinson’s disease (PD), and Huntington’s disease (HD) [52,53]. Interestingly, transcription factors such as Nrf2 govern the homeostatic oxidative responses, highlighting its potential as a therapeutic target for counterbalancing oxidative damage during neurodegenerative disorders. Indeed, Nrf2 has been shown to upregulate the expression of various antioxidant enzymes, which can be further stimulated with CAPE. Next, we will briefly provide mechanistic insights on the effects of chemical modifications during OS associated with neurodegeneration. Further, we will examine evidence supporting the potential of CAPE for positively impacting Nrf2 signaling during neurodegeneration [54,55].

Metabolic activity is a major source of OS in the cells. As such, mitochondria are the major sources of cellular ATP and the main generators of ROS and OS inside the cells. Indeed, around 2% of the oxygen consumed by cells is converted into ROS during oxidative phosphorylation in the mitochondria. Different mechanisms may account for the tight association of OS with neurodegeneration. For instance, mitochondrial exposure to ROS may cause damage of its DNA (DNAmit), including DNA fragmentation, chromatid exchange, translocations, and consequent mutations. Altered DNAmit impacts the expression and abnormal activity of proteins involved in ATP synthesis by uncoupling the electronic transport chain from oxidative phosphorylation [54,55], decreasing ATP production and therefore the activity of the plasma membrane ATPase. Decreased activity of the sodium/potassium transporter (Na/K-ATPase) leads to neuronal membrane depolarization, thus generating an increase in cellular excitation and facilitating the opening of voltage-sensitive Ca^2+^ channels. In turn, increased glutamate release (and decreased uptake by the glutamate transporter) leads to activation of the N-methyl-D-aspartate (NMDA)-glutamate receptor, thus contributing to increased Ca^2+^ influx [55,56]. Moreover, calcium activates several Ca^2+^-dependent signaling pathways, such as nitric oxide synthases (NOS), leading to an excess in the production of nitric oxide (NO), which also generates free radicals causing oxidative damage [57,58,59,60]. 

Oxidative damage is manifested as covalent modifications of different subcellular structures and compartments. Accordingly, ROS can oxidate macromolecules of various kinds, including lipids, proteins and nucleic acids, modifying their structure and function [61,62]. As such, lipid damage by peroxidation is associated with early and late markers, including hexanonyl-lysine (HEL), acrolein-lysine (ACR), malondialdehyde, F2-isoprostanes and 4-hydroxynonenal (4-HNE) adducts. In turn, oxidative damage of DNA and RNA is reflected in markers such as 8-hydroxy-2-deoxyguanosine (8-OHdG) and 8-hydroxyguanine (8-oxo-dGo), respectively, which have a high mutagenic power [53,63]. For AD and HD, an increase of oxidized molecules such as proteins, lipids and nucleic acids is generated by the presence of 3-nitrotyrosine and 3,3’-dithyrosine in the hippocampus, neocortex and cerebrospinal ventricular fluid of patients with the condition [64]. Furthermore, nuclear and mitochondrial DNA modification and lipoperoxidation are associated with increased 8-OHdG, acrolein, malondialdehyde, 4-HNE and F2-isoprostanes [65]. It has been shown that there is a relationship between increased 4-HNE in cultured hippocampal cells and altered Na+/K+ ATPase function [66,67]. PD patients show a reduced activity of antioxidant enzymes such as catalase and glutathione peroxidase, and the most affected dopaminergic brain region presents a decrease in glutathione (GSH) levels, so the ratio of GSH/GSSG (oxidized glutathione) decreases during neuronal degeneration, which favors the formation of free radicals [68,69]. Moreover, as in amyotrophic lateral sclerosis (ALS), there is a decrease in the activity of Complex I of the mitochondrial electron transport chain accompanied by an increase of up to 7 fold in O_2_, H_2_O_2_ and NO production [6].

The transcription factor Nrf2 is the intrinsic controller of cellular redox homeostasis and regulates the expression of a set of antioxidant genes called phase II genes, which produce phase II proteins associated to ROS stabilization. Under normal redox conditions, Nrf2 associates with the Keap1 protein, forming the Keap1–Nrf2 complex that is retained in the cytosol (cytoskeleton) [8,9,10]. Nevertheless, in an oxidizing environment, phosphatidylinositol 3-kinase (PI3K) produces depolymerization of actin microfilaments, realizing the Nrf2 from the cytoskeleton [70], facilitating its nuclear translocation, allowing its binding to a specific DNA sequence known as ARE (Antioxidant Response Element). PI3K furthermore produces enzymes of phase II as NQO1 (NADPH Quinone Dehydrogenase 1), glutathione S-transferase (GST) that conjugates ROS and hydrophobic electrophiles to glutathione, UDP-glucuronosyl transferases that conjugate xenobiotics to glucuronic acid for excretion, ferritin for iron storage, epoxide hydrolase that inactivates epoxides, and γ-glutamyl cysteine synthetase (γ-GCS) involved in GSH synthesis, heme oxygenase 1 (HO-1) that catabolizes heme groups, as well as exerting antioxidant and anti-inflammatory actions [71,72].

CAPE shows potent antioxidant activity, reducing the formation of the superoxide anion produced during the autoxidation of β-mercaptoethanol and quenching the 2,2-diphenyl-1-picrylhydrazyl (DPPH) radical. It also inhibits the activity of xanthine oxidase (XO). In addition, CAPE exhibits potent cytoprotective and antigenotoxic antilipoperoxidative potential against oxidative damage [73,74,75]. CAPE activates the expression of the antioxidant HO-1 and protects the kidneys from aging-related oxidative injury in rats by reducing malondialdehyde (MDA) levels with a simultaneous elevation of superoxide dismutase, catalase, glutathione peroxidase, and reduced glutathione in the renal tissues of old rats [76,77]. CAPE possesses a marked iron-binding capacity and therefore interferes with the formation of the ferrous-ferrozin complex [78]. Sun et al. demonstrated that at micromolar concentrations in a concentration-dependent manner, CAPE is capable of increasing Nrf2 by inhibiting ubiquitination and promoting Nrf2 with subsequent HO-1 expression [79]. Furthermore, Sun et al. demonstrated that CAPE regulates NF-κB signaling and the inflammatory response through the Nrf2/HO-1 signaling pathway in IL-1β-stimulated chondrocytes [80]. In HepG2 cells, CAPE induces accumulation of Nrf2 in the nucleus with the expression of HO-1 regulated by Kelch-like ECH-associated protein 1, where the oxidized state of the catechol moiety of CAPE activates the via more potently due to oxidized CAPE binds to keap-1, releasing the Nrf2 factor and allowing its nuclear translocation [81].

CAPE attenuates oxidative stress and neurotoxicity by modulating MAPK and Akt/glycogen synthase kinase 3β (GSK3β)-signaling pathways and reducing dead hippocampal cells in an AD mouse model. CAPE thus is particularly beneficial in AD treatment because the disease is closely related to toxicity in the hippocampus. Moreover, CAPE was found to protect transient ischemic injury and prevent neonatal encephalopathy by blocking NF-κB-mediated neuronal inflammation, evoking antioxidant effects and mitochondrial apoptosis [82]. 

CAPE may be able to ameliorate Parkinson’s disease in PD murine models, attenuating dopaminergic loss and decreasing behavioral abnormality through oxidative stress and reducing neuroinflammation, causing a protective effect on nigral dopaminergic neurons by HO-1-dependent MAPK signaling [83,84]. 

In an acute brain damage model of stroke, CAPE decreased MDA levels, prevented lipid peroxidation due to ROS scavenging, and increased GSH levels and NO in brain tissue [85]. When CAPE was compared with alpha-tocopherol, a potent free radical scavenger molecule (Vitamin E) on ischemia-reperfusion brain injury, the former proved to be better than Vitamin E in reducing the brain level of MDA, a marker of oxidative stress [86]. Moreover, CAPE is highly effective in scavenging free radicals, even more than the structurally related ferulic acid and ethyl ferulate, as well as in blocking the enhanced release of cardiolipin and cytochrome c. Therefore it protects from functional alterations in isolated mouse brain mitochondria challenged by anoxia/reoxygenation [87]. This compound further exhibits an effect on isolated brain mitochondria by directly inhibiting Ca^2+^-induced cytochrome c release, caspase-3, caspase-1, and reducing the expression of inducible nitric oxide synthase by hypoxia-mediated mechanisms, showing neuroprotection activity in vivo. Moreover, CAPE potently blocked nitric oxide-induced neurotoxicity in vitro [88]. The mechanisms of CAPE on oxidative stress and activation of the Nrf2 pathway are summarized in Figure 8.

## 6. CAPE Inhibits Neuroinflammation by Modulation of the NF-κB Pathway 

Cumulating evidence supports that NF-κB plays a critical role in regulating the immune response and inflammation implicated in the pathogenesis of neurodegenerative disorders. The inflammatory response involves the activation of the innate and adaptive immune system, leading to the release of proinflammatory and anti-inflammatory cytokines, which play a crucial role in resolving inflammation. These systemic inflammatory responses, known as neuroinflammation, are also manifested inside the brain’s parenchyma. Furthermore, there is a particular inflammatory signature in the bloodstreams and brains of patients with different neurodegenerative disorders, including AD, PD, ALS, and FTD. Interestingly, CAPE may be a relevant therapeutic agent for controlling NF-κB signaling and reducing neuroinflammation in these disorders. 

NF-κB activation can be mediated through three pathways: canonical (or classical), the non-canonical (or alternative), and a non-enzymatic regulatory component. In the canonical pathway, NF-κB is activated in response to viral and microbial infections or exposure to pro-inflammatory cytokines. This occurs with the participation of RelA and c-Rel dimers together with P50. After activation, the IKKB subunit phosphorylates IkB at serine residues, causing its ubiquitin-dependent degradation via proteasome, translocating NF-κB (p65/p50) to the nucleus to act as a transcription factor for the target gene [89]. The alternative pathway is activated via members of the TNF-α family, such as lymphotoxin B and BAFF; CK2 is independent of the IKK complex (IKKα and IKKβ). The pathway mainly activates IKKa, which is responsible for the phosphorylation of p100, causing proteolysis to take an active form, which is p52. The dimers can then be translocated into the nucleus, acting as transcription factors [90].

Most of the genes under NF-κB transcriptional control are involved in immune system signaling and inflammatory responses. The transcriptional control of cytokine expression by NF-κB is probably one of the most important factors the pathogenic role of NF-κB in some diseases is assessed. Some of these cytokines include TNFα, IL-1α/β, IL-2, 3, 6, 12, GM-CSF, M-CSF and G-CSF, the chemokines MCP-1, KC, MIP-1 and CCL, together with adhesion molecules ICAM-1, E-selectin and VCAM-1, which enable the recruitment and binding of immune system cells to sites where the inflammatory process develops [91]. NF-κB prevents apoptosis in several inflammatory cells, including macrophages, dendritic cells, T and B-lymphocytes, and neutrophils, and promotes survival of several malignant tumors, especially of lymphomas. In contrast, the inflammatory response can induce apoptotic cell death. This inflammatory cell death response is initiated by the production of cell death receptors (Fas and FasL) and intracellular apoptosis-inducing proteins. This apoptotic death is further assisted by activated immune cells, which secrete granzyme, perforin and nitric oxide, all apoptosis-inducing factors regulated by NF-κB [91]. 

The mammalian NF-κB family consists of five subfamilies: NF-κB1 (p105 and p50), NF-κB2 (p100 and p52), RelA (p65), RelB, and c-Rel. In most cell types, NF-κB dimers are predominantly cytoplasmic because they are in constant interaction with NF-κB inhibitors (IkBα, IkBβ, IkBε and Bcl-3) remaining transcriptionally inactive [92]. IkBα, IkBβ, and IkBε interact with NF-κB dimers and are responsible for their retention in the cytoplasm. These NF-κB inhibitors contain conserved serine residues that can be phosphorylated by IkB kinases (or IKKs) to be subsequently degraded by the proteasome and finally generate the translocation of NF-κB dimers to the nucleus [80,93].

In 1996, the molecular mechanisms by which caffeic acid derivatives prevent NF-κB activation were analyzed for the first time. Initially, it was shown that the activation of NF-κB by a tumor necrosis factor (TNF) is completely blocked by CAPE in a dose- and time-dependent manner. In addition to TNF, CAPE also inhibits NF-κB activation induced by other inflammatory agents, including phorbol ester, ceramide, hydrogen peroxide, and okadaic acid. However, reducing agents reverse the inhibitory effects exerted by CAPE, suggesting a role for critical sulfhydryl groups in NF-κB activation [94]. CAPE prevents translocation of the p65 subunit of NF-κB to the nucleus but has no significant effect on TNF-induced degradation of IκBα; however, it delays IκBα resynthesis. Therefore, these data suggest that CAPE prevented NF-κB activation by inhibiting p65 translocation and IκBα degradation in human chondrocytes [80]. In HCT116 cells, CAPE inhibited NF-κB by direct inhibition of IKK [95].

The effect of CAPE on the inhibition of NF-κB is to binding specific target-DNA sites, whereas other transcription factors including AP-1, Oct-1, and TFIID, were found to be unaffected by CAPE [94]. When several synthetic structural analogs of CAPE were tested, it was found that a bicyclic, rotationally restricted, 5,6-dihydroxy form exhibits high inhibitory activity, whereas the 6,7-dihydroxy variant is less active. In this sense, other investigations in which caffeic acid phenethyl ester (CAPE) and 3,4-dihydroxybenzalacetone (DBL), both phenylpropanoid derivatives containing catechol with and presenting diverse bioactivities, have been evaluated. It was determined that the proinflammatory effects on NO synthase expression were strongly suppressed by CAPE and, to a lesser extent, by DBL and caffeic acid ethyl ester. Thus, the induction of genes downstream of LPS-activated NF-κB, such as NO synthase, IL-1β and IL6, and the translocation of NF-κB p65 to the nucleus were reduced, to a greater extent by CAPE than by DBL. Interestingly, phosphorylation of p65 was reduced by both compounds, especially by CAPE, even when the levels of IκB were not altered. One explanation for the action of these compounds is related to the effect of CAPE and DBL on the thiol groups of p65, which were modified by these molecules. However, the inhibitory effects on p65 phosphorylation and nitrite production were reversed by pretreatment with thiol-containing reagents, suggesting that CAPE has strong inhibitory effects on NF-κB activation due to the modification of thiol groups and the phosphorylation of p65 [94].

In activated astroglia cells, CAPE pretreatment abrogated TNF-α-induced expression of chemokine (CC motif) ligand 2 (CCL-2) and intercellular adhesion molecule 1 (ICAM-1) through the inhibition of NF-κB activation in a cell-specific manner [96]. In another study, CAPE was found to mediate up-regulation of Abcd2 expression and peroxisomal β-oxidation. This causes a decrease in the very long chain fatty acid (VLCFA) levels in ABCD1-deficient U87 cells; VLCFA accumulation is characteristic of a neuroinflammatory disease associated with demyelination of the cerebral white matter known as: X-linked adrenoleukodystrophy (X-ALD). CAPE reduces NF-κB activation, iNOS expression. and inflammatory cytokines in primary cultures of astrocytes derived from ABCD1/ABCD2- silenced mice [97].

Microglial activation mediates inflammatory processes crucial in the development of neurodegenerative disorders. CAPE proved to inhibit cyclooxygenase-2 (COX-2) and subsequent NO production for both in vitro and in vivo models of microglial activation, mediated by α-adenosine monophosphate-activated protein kinase 5′ (AMPK), erythropoietin (EPO), and HO-1 [41]. It is known that with Amyotrophic Lateral Sclerosis, the mutation of superoxide dismutase (SOD) is highly correlated with the pathogenesis of the disease. A study using a murine model showed that CAPE prolongs the survival of mice expressing mutant SOD1 (G93A) by attenuating neuroinflammation and motor-neuron cell death as a consequence of p38 phosphorylation reduction [98]. CAPE may exert its anti-inflammatory effects by inhibiting ROS production at the transcriptional level, suppressing NF-κB activation, and directly inhibiting the catalytic activity of iNOS [99]. Supposed mechanisms of CAPE actions through the NF-κB pathway are summarized in Figure 9.

## 7. Biological Properties of CAPE

Numerous studies have evidenced the medicinal properties of propolis, considering it a superfood. Many of these beneficial effects are attributable to CAPE. See Table 2.

The low solubility of CAPE and therefore its low bioavailability limit its therapeutical use. These problems can be overcome by packing the drug in nanoparticles or vesicles, which provide better therapeutic effects at lower doses. Table 3 summarizes the main delivery systems that have been used to enhance the pharmacological effects of CAPE.

## 8. Conclusions

Nowadays neurodegenerative diseases are a global concern with a dramatically rapid increase. Existing treatments are still palliatives. In this context, functional food could be an alternative in the prevention or deceleration of neurodegenerative problems. For future treatment of AD, propolis has proven to potentiate the neuroprotective effect of memantine in preclinical models [133]. This evidence supports a new potential avenue of investigation of a combination therapy of CAPE with other drugs to modify the evolution of AD. CAPE has a wide range of pharmacological activities, including neuroprotective effects by inhibiting NF-κB pathways and evoking intrinsic cell protection by increasing the Nrf2 pathway. The main objective of this review is to provide a comprehensive overview of the therapeutic potential of CAPE in the treatment of neurodegenerative disorders through reviewing the published data about its neuroprotective effects in the last decade. We also shed light on its molecular mechanisms of action and summarize the structure--activity relationship of synthetic derivatives of CAPE, which will be useful for the further design and development of better derivatives.

## Figures and Tables

**Figure 1 antioxidants-12-01500-f001:**
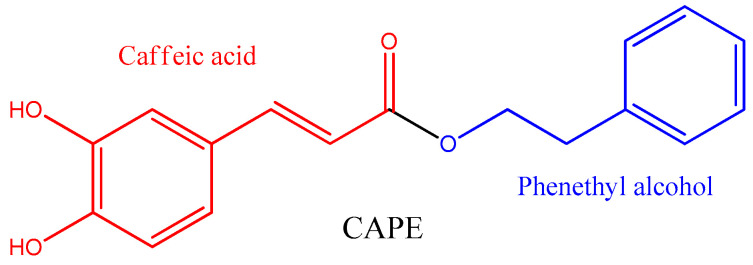
Chemical structure of caffeic acid phenethyl ester (CAPE).

**Figure 2 antioxidants-12-01500-f002:**
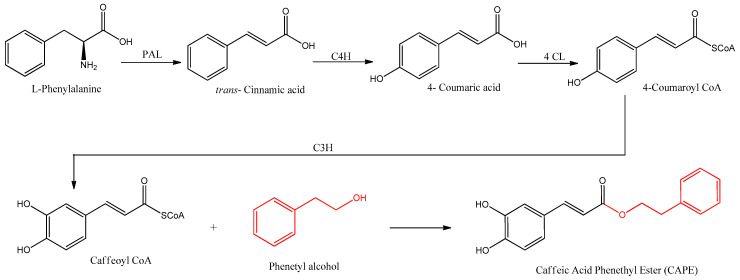
Biosynthetic pathways for the formation of CAPE. The involved enzymes are abbreviated: PAL = phenylalanine ammonia lyase; C4H = cinnamate 4-hydroxilase; 4CL = 4-coumaric acid CoA-ligase; C3H = p-coumarate 3-hydroxylase.

**Figure 3 antioxidants-12-01500-f003:**
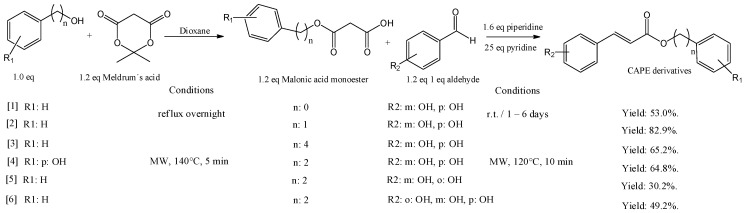
Synthesis of CAPE and derivatives by Knoevenagel condensation using thermal or microwave methods.

**Figure 4 antioxidants-12-01500-f004:**
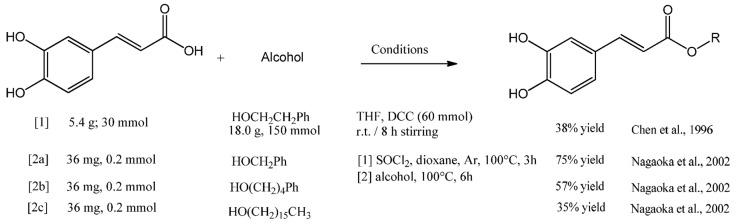
Synthesis of CAPE derivatives by DCC and acyl chlorine condensation [32,39].

**Figure 5 antioxidants-12-01500-f005:**
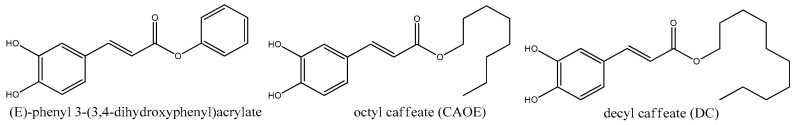
Chemical structure of caffeoyl ester derivatives.

**Figure 6 antioxidants-12-01500-f006:**
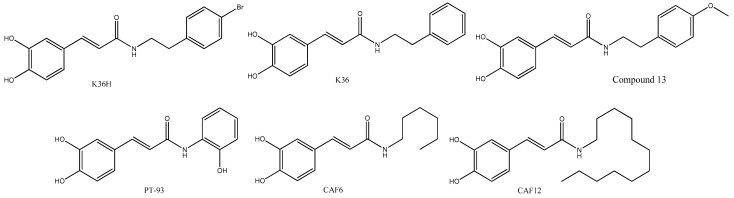
Chemical structure of caffeoyl amide derivatives.

**Figure 7 antioxidants-12-01500-f007:**
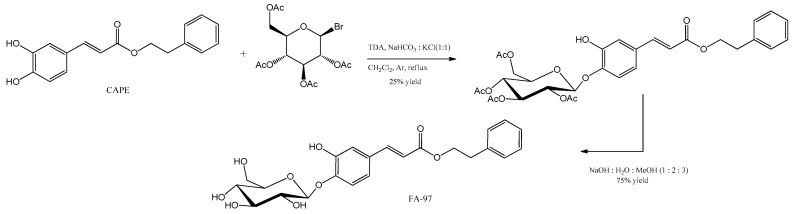
Synthesis of caffeic acid phenethyl ester 4-O-glucoside, or FA-97 [51].

**Figure 8 antioxidants-12-01500-f008:**
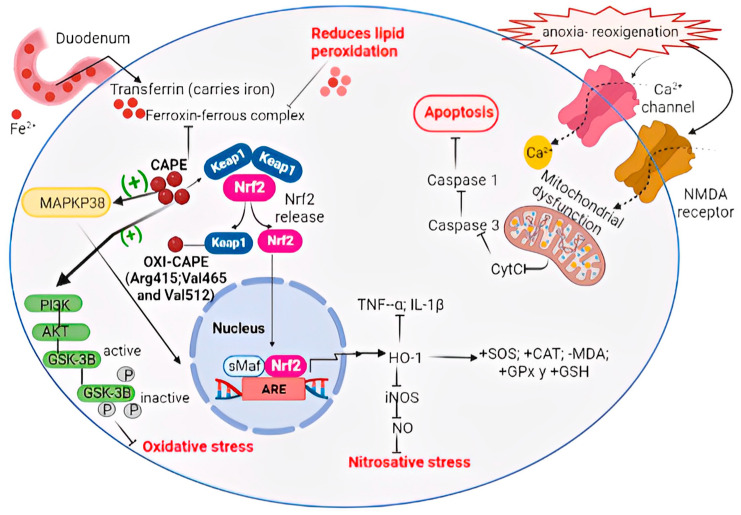
CAPE action mechanisms on oxidative stress and activation of the Nrf2 pathway. (-) routes that inactivate. (+) activating routes.

**Figure 9 antioxidants-12-01500-f009:**
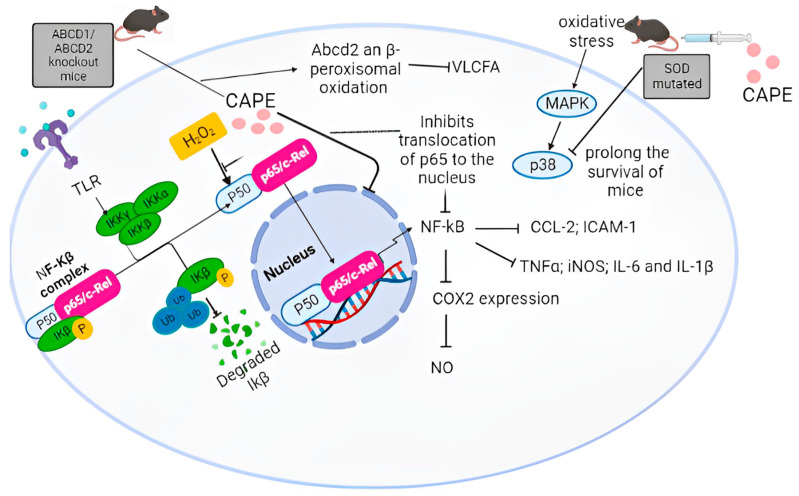
Mechanism of action of CAPE against neuroinflammation and NF-κB pathway activation. (-) routes that inactivate. (+) activating routes.

**Table 1 antioxidants-12-01500-t001:** Concentration of CAPE in different natural sources.

Source	Concentration of CAPE (mg g^−1^)	Reference
*Populus buds*	1.611 ± 0.272	[25]
Propolis from Mexico (Ures)	11.4	[26]
Propolis from Spain (Viveros)	10.1	[27]
Propolis from Spain (Valencia)	10.4	[27]
Propolis from Chile	5.8	[27]
Propolis from China	4.9	[27]

**Table 2 antioxidants-12-01500-t002:** Biological properties of CAPE.

Pharmacological Effect	Molecular Mechanism	Reference
Wound repair	CAPE promotes early inflammatory response (increased NOS2, TNF-α, and NF-κB) associated with a short-term event, leads to fast skin-wound healing and inhibition of inflammation. Significant increase in the glutathione (GSH) level, an endogenous antioxidant that plays a key role in cellular defense against oxidative stress. Considerable decrease in malondialdehyde (MDA) and superoxide dismutase activity.	[100,101]
Antidiabetic properties	CAPE is a heme oxygenase 1 (HO-1) inducer, combating the increase of reactive oxygen species (ROS) induced by hyperglycemia. It inhibits the 5-lipoxygenase, alleviating diabetic atherosclerotic symptoms (an important macrovascular complication of diabetes). It restores adipocyte function by increasing adiponectin and PPARγ (their activation leads to improved insulin sensitivity), leading to the reduction of proinflammatory factors.	[102,103,104,105]
Anticancer properties	Inhibition of DNA synthesis, disruption of growth signal transmission, induction of apoptosis through an internal apoptotic pathway, and promotion of anti-angiogenic effects. It enhances the anti-cancer effect of first-line chemotherapy drugs, an example being the drug paclitaxel in a rat model of DMBA-induced breast cancer, resulting in lower tumor weights compared to those with paclitaxel alone. CAPE protects normal cells against the effects of anticancer drugs by acting as a chemopreventive agent. The drug irinotecan protects normal blood, liver, and kidney cells without affecting the cytotoxicity of irinotecan in the in vivo model of Ehrlich ascites tumor cells.	[94,106,107,108,109]
Periodontitis Treatment	It improves bone healing, preventing RANKL-induced osteoclastogenesis (causes follicular bone destruction), suggesting its use as a regenerative agent in therapy of bone resorption.	[110,111]
Antibacterial activity	High activity against *Mycobactrium tuberculosis, Mycobacterium avium, Streptococcus piogenes Klebsiella pneumoniae* and *Staphylococcus epidermidis*. It presents synergistic activity with many antituberculosis drugs, including rifamycin, streptomycin, and isoniazid, and antibiotics such as gentamicin, tetracycline, chloramphenicol, vancomycin, clindamycin, netilmicin. CAPE inhibits biofilm formation and lactic acid and extracellular polysaccharide production in *S. mutans.*	[112,113,114,115]
Antiatherogenic effects	In human platelets, CAPE (15 and 25 µM) markedly inhibits platelet aggregation stimulated by collagen (2 µg/mL).	[116]
Estrogenic effects	CAPE is a selective agonist of ER-β estrogen receptors (present in the lungs, blood vessels, brain, and bones).	[117]
Hepatoprotective effects	In the liver, CAPE elevates tissue catalase (CAT) activity and ameliorates ultrastructural changes associated with aging; it also gives protection against necrosis, lipid peroxidation, aberrant cell proliferation, and p65 activation (decreased number of preneoplastic nodules and reduced incidence of liver tumors). Reduced malondialdehyde (MDA) levels and increased activities of glutathione peroxidase (GPx), together with superoxide dismutase (SOD) in liver tissue.	[77,118]
Lung protection	Effects on pulmonary arterial hypertension and fibrosis by HIF-1α/platelet-derived growth factor (PDGF)-dependent Akt/ERK pathways.	[119]
Protection of the bone diseases	CAPE induces osteoclast apoptosis, antioxidant effects and the modulation of osteoprotegrin signaling pathways. By suppressing NF-κB activity, CAPE significantly inhibits osteoclastogenesis and osteoclast differentiation.	[120,121,122]
Alzheimer’s disease	CAPE significantly inhibits neuronal apoptosis and neuroinflammation, induces Nrf2 activation, inhibits glycogen synthase kinase 3β in the hippocampus, and improves learning and memory and cognition.	[123]

**Table 3 antioxidants-12-01500-t003:** Delivery systems for novel CAPE formulations.

Formulation	Target	Results	Reference
Poly(lactic-co-glycolic acid) (PLGA) co-loaded nanoparticles (QuCaNP) quercetin and CAPE	Improving anticancer efficacy in HT-29 human colorectal carcinoma.	Increased caspase-3 (2.38-fold) and caspase-9 (2-fold) mRNA levels and expressions of key proteins in the intrinsic apoptosis pathway in HT-29 cells.	[124]
Copolymer: polyglycerol and poly(allyl glycidyl ether) (C12-PAGE-PG), loaded with CAPE	To evaluate the in vitro and in vivo safety of a CAPE-loaded micellar system as a drug delivery platform on HepG2 cells.	Empty micelles loaded with CAPE showed no cytotoxic effects and retained the cytotoxic activity of CAPE loaded in the micelles, making it a good strategy to use this hydrophobic compound and improve the effectiveness of the treatments.	[125]
NiO nanoparticles and MnO_2_/NiO nanocomposites with guanidine and CAPE as a carrier	Evaluate its capacity as an anchoring method for drug carriers.	The drug loading time was 100 min and drug release in 1–10 h with 20–80% drug release.	[126]
Hyaluronic acid (HA) conjugated with phenylboronic acid pinacol ester (PBPE) with radiosensitive delivery of CAPE	Manufacture radiosensitive delivery of CAPE for application in radioprotection.	Prevention of radiation-induced apoptosis and intracellular ROS accumulation, with increased survivability of mice against radiation-induced death.	[127]
Methoxy poly(ethylene glycol)*-b* -poly(ε-caprolactone) copolymer nanoparticles (CE) with CAPE.	Study antitumor activity against lung metastasis by CT26 colon carcinoma cells.	Superior anti-metastatic efficacy against the tumor than CAPE itself.	[128]
Caffeic acid phenethyl ester-morphthalin antibody nanoparticles	Generate a potent anticancer drug by recruiting an anti-mortalin antibody (hsp70 chaperone that is enriched on the cancer cell surface).	Enhanced growth arrest/apoptosis of cancer cells through down-regulation of cyclin D1-CDK4, phospho-Rb, PARP-1 and the anti-apoptotic protein Bcl2. Significantly increased expression of p53, p21 WAF1, and caspase cleavage. Significantly improved down-regulation of proteins involved in cell migration.	[129]
Folic acid-conjugated PLGA nanoparticles.	Improve cytotoxicity, solubility, and achieve sustained release of CAPE.	It showed enhanced cytotoxicity in vivo and in vitro, causing a decrease in cell proliferation by 46%.	[130]
Stimuli-responsive liposomal nanocarrier loaded with CAPE modified with ac peptide (RGDyK)	Attack ischemic lesions and remodel neurovascular units (NVU) to reduce the progression of brain injury.	Drugs release in response to pathological signaling stimuli, localization of cerebral ischemia-reperfusion injury, and remodeling of neurovascular units by reducing neuronal apoptosis, regulating microglia polarization and repair of vascular endothelial cells.	[131]
Nano-Liposomal Formulation of CAPE	New strategies for acute pancreatitis treatment, evaluated in rat model	CAPE-loaded-NL showed better antioxidant, anti-inflammatory, and anti-apoptotic effects than free CAPE	[132]

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
