# Peer review of "Caffeic Acid Phenethyl Ester (CAPE): Biosynthesis, Derivatives and Formulations with Neuroprotective Activities"

_antioxidants, 2023, doi:10.3390/antiox12081500_

Round 1

Reviewer 1 Report

This manuscript reviewed the biosynthesis and chemical synthesis of CAPE derivatives, their miscellaneous activities, and their neuroprotective activity related to Nrf2 and NF-κB pathways.

A few concerns for the authors.

1.       Page 2, line 70-71, “Nuclear erythroid transcription factor 2 (Nrf2) responds to oxidative stress”, page 3, line 123; and page 8, line 232, Nrf2 had the full name before in the text, no need to provide the full name again. Similarly, page 3, line 124, no need to provide again the full name for NF-kB. Also need to pay attention to some other abbreviations to make sure their full names do not show up too often.

2.       Page 2, line 79-82, “For instance, in the pharmacological treatment of AD, since the 1990s, only tree cholinesterase inhibitors including galantamine, rivastigmine, and donepezil together with memantine a glutamatergic antagonist are available.” Is not complete. In 2021 and 2023, FDA has approved aducanumab and lecanemab, both of which are Ab antibodies.

3.       Page 4, the authors discussed some of the PK properties of CAPE but not much about the metabolism of this compound. It would make more sense to provide the metabolism information for CAPE.

4.        In session 5. CAPE inhibits oxidative stress by modulation of the Nrf2 pathway, no detailed information about how CAPE modulates the Nrf2 pathway has been clearly presented. Information about what is the protein target(s) for CAPE to enhance Nrf2 pathway, how targeting this (these) protein target(s) activate Nrf2 pathway need to be provided.

5.       In session 6. CAPE inhibits neuroinflammation by modulation of the NF-κB pathway, again more detailed mechanism needs to be provided regarding how CAPE prevent the activation of NF-κB pathway. What are the protein targets and how the interaction between CAPE and its protein target leads to inhibition of NF-κB pathway.

6.  Besides low solubility, CAPE should also suffer from chemical instability as the catechol hydroxy groups are susceptible to oxidation. Any structure modifications or formulations to increase its stability?

Author Response

Review - antioxidants-2447108

Caffeic acid phenethyl ester (CAPE): Biosynthesis, derivatives and formulations with neuroprotective activities

Rebeca Perez , Viviana Burgos , Víctor Marín , Antonio Camins , Jordi Olloquequi. , Ivan Gonzalez-Chavarría , Henning Ulrich , Ursula Wyneken , Leandro Ortiz , Cristian Paz *

Comments and Suggestions for Authors, Reviewer 1.

  1. Page 2, line 70-71, “Nuclear erythroid transcription factor 2 (Nrf2) responds to oxidative stress”, page 3, line 123; and page 8, line 232, Nrf2 had the full name before in the text, no need to provide the full name again. Similarly, page 3, line 124, no need to provide again the full name for NF-kB. Also need to pay attention to some other abbreviations to make sure their full names do not show up too often.

Response: checked

  1. Page 2, line 79-82, “For instance, in the pharmacological treatment of AD, since the 1990s, only tree cholinesterase inhibitors including galantamine, rivastigmine, and donepezil together with memantine a glutamatergic antagonist are available.” Is not complete. In 2021 and 2023, FDA has approved aducanumab and lecanemab, both of which are Ab antibodies.

Response: we changed the sentence to:

For instance, in the pharmacological treatment of AD, since the 1990s, only tree small molecules as cholinesterase inhibitors including galantamine, rivastigmine, and donepezil together with memantine a glutamatergic antagonist are available, together two antibodies aducanumab and lecanemab.

  1. Page 4, the authors discussed some of the PK properties of CAPE but not much about the metabolism of this compound. It would make more sense to provide the metabolism information for CAPE.

Response: we added a new reference and include an additional paragraph, trying to explain better this point.

  1. In session 5. CAPE inhibits oxidative stress by modulation of the Nrf2 pathway, no detailed information about how CAPE modulates the Nrf2 pathway has been clearly presented. Information about what is the protein target(s) for CAPE to enhance Nrf2 pathway, how targeting this (these) protein target(s) activate Nrf2 pathway need to be provided.

Response: We added a new reference (81) but the mechanism of interaction between CAPE or oxidized cape with the Keap-1 is not fully elucidated

  1. In session 6. CAPE inhibits neuroinflammation by modulation of the NF-κB pathway, again more detailed mechanism needs to be provided regarding how CAPE prevent the activation of NF-κB pathway. What are the protein targets and how the interaction between CAPE and its protein target leads to inhibition of NF-κB pathway.

Response: we added a new reference (95), complementing the information about the interaction of CAPE with IKK, but the specific site of binding or interaction CAPE-IKK or with IκBα is not fully understand.

  1. Besides low solubility, CAPE should also suffer from chemical instability as the catechol hydroxy groups are susceptible to oxidation. Any structure modifications or formulations to increase its stability?

Response: we did not find a formulation than evaluate molecular stability, just we found a new formulation than improves activity by Nf-kB and Nrf2, that was added to the references.

Shahin, N. N., Shamma, R. N., & Ahmed, I. S. (2022). A nano-liposomal formulation of caffeic acid phenethyl ester modulates Nrf2 and NF-κβ signaling and alleviates experimentally induced acute pancreatitis in a rat model. Antioxidants, 11(8), 1536.

Reviewer 2 Report

1. the reference in Table 1 needs to be indicated according to mdpi guidelines []

2. the material and methods section is not present, please add it. 

3. Considering that the review was focused on the biological activity of CAPE, a natural bioactive compound, a section or a table related to the possible natural CAPE food/plant sources is highly suggested. 

4. considering that all the biological activity investigated i,e., antioxidant ad antiinflammatory are generally described in the brain system, in my opinion, the title could be changed because the CAPE application studied is related to its neuroprotective activity, which includes both anti-inflammatory and antioxidant activity. 

5. why the author have added the paragraph "3. Biological properties of CAPE" when in the following section each molecular mechanism is deepened? 

Author Response

Caffeic acid phenethyl ester (CAPE): Biosynthesis, derivatives and formulations with neuroprotective activities

Rebeca Perez , Viviana Burgos , Víctor Marín , Antonio Camins , Jordi Olloquequi. , Ivan Gonzalez-Chavarría , Henning Ulrich , Ursula Wyneken , Leandro Ortiz , Cristian Paz *

Comments and Suggestions for Authors, Reviewer 2.

  1. the reference in Table 1 needs to be indicated according to mdpi guidelines [ ]

Response: the references were changed to the journal guidelines

  1. the material and methods section is not present, please add it. 

Response: Dear reviewer we do not use to write material and methods in a review, also other reviews published in the journal Antioxidants don´t have this section, please, we would like to follow this idea.

  1. Considering that the review was focused on the biological activity of CAPE, a natural bioactive compound, a section or a table related to the possible natural CAPE food/plant sources is highly suggested. 

Response: We added a table (Table 1) with that information.  

  1. considering that all the biological activity investigated i,e., antioxidant ad antiinflammatory are generally described in the brain system, in my opinion, the title could be changed because the CAPE application studied is related to its neuroprotective activity, which includes both anti-inflammatory and antioxidant activity. 

Response: considering your suggestions we change the title to: caffeic acid phenethyl ester (CAPE): Biosynthesis, derivatives and formulations with neuroprotective activities.

  1. why the author have added the paragraph "3. Biological properties of CAPE" when in the following section each molecular mechanism is deepened? 

Response: We think that this information is valuable, for that reason we moved the chapter 3 (before) to chapter 7, now is more coherent both information, one is the miscellaneous activity of CAPE and second are the new formulations that improve the bioactivity of CAPE